# Application and Research of IoT Architecture for End-Net-Cloud Edge Computing

Yongqiang Zhang [1,2,3], Hongchang Yu [1], Wanzhen Zhou [1,3,*] and Menghua Man [2,*]

1 School of Information Science and Engineering, Hebei University of Science and Technology, Shijiazhuang 050018, China
2 National Key Laboratory on Electromagnetic Environment Effects, Army Engineering University of PLA, Shijiazhuang Campus, Shijiazhuang 050003, China
3 Hebei Technology Innovation Centre of Intelligent IoT, Shijiazhuang 050018, China
* Correspondence: zhou_wanzhen@163.com (W.Z.); manmenghua@126.com (M.M.)

**Abstract:** At the edge of the network close to the source of the data, edge computing deploys computing, storage and other capabilities to provide intelligent services in close proximity and offers low bandwidth consumption, low latency and high security. It satisfies the requirements of transmission bandwidth, real-time and security for Internet of Things (IoT) application scenarios. Based on the IoT architecture, an IoT edge computing (EC-IoT) reference architecture is proposed, which contained three layers: The end edge, the network edge and the cloud edge. Furthermore, the key technologies of the application of artificial intelligence (AI) technology in the EC-IoT reference architecture is analyzed. Platforms for different EC-IoT reference architecture edge locations are classified by comparing IoT edge computing platforms. On the basis of EC-IoT reference architecture, an industrial Internet of Things (IIoT) edge computing solution, an Internet of Vehicles (IoV) edge computing architecture and a reference architecture of the IoT edge gateway-based smart home are proposed. Finally, the trends and challenges of EC-IoT are examined, and the EC-IoT architecture will have very promising applications.

**Keywords:** Internet of Things (IoT); industrial Internet of Things (IIoT); cloud computing; edge computing





## 1. Introduction

The development of IoT technology makes it possible for everything to be connected. Cloud computing [1] simplifies the process of IoT data collection, processing and storage through powerful data processing and storage capabilities. It provides elastic and scalable infrastructure services such as computing, storage and networking for business applications. This traditional cloud computing platform adopts a centralized architecture for non-real-time, long-cycle data and business decision scenarios with high reliability and on-demand distribution.

However, with the popularization and deployment of 5th-Generation (5G) mobile technology, the volume of IoT edge devices and data have increased dramatically [2]. According to International Business Machines Corporation (IBM), the number of IoT edge devices is expected to reach approximately 55 billion in 2022 and 150 billion by 2025, with endpoint data volumes reaching 300 ZB. As shown in Table 1, cloud computing technologies in traditional IoT architectures have emerged as limitations in terms of centralized data processing models and technology development in meeting the new specific requirements of a broader range of IoT scenarios in the 5G era. There is a significant emphasis on image, video, data recognition and processing skills or highly demanding needs for low latency and high network bandwidth, especially in emerging IoT application areas such as autonomous driving [3], drones [4], smart homes [5] and smart cities [6]. Therefore, extending capabilities such as computation, control, storage and services from the network core to the network edge to optimize the IoT architecture is a robust research solution that

promises to solve the above problems, called edge computing [7]. Edge computing bypasses bandwidth and latency bottlenecks by evolving the centralized cloud computing model in traditional IoT architectures into a center-region-edge distributed computing model, resulting in an IoT edge computing (EC-IoT) architecture with collaborative computing between edge and cloud computing. Compared to traditional IoT architectures, this architecture will support faster and more comprehensive data analysis, gain deeper insights, reduce response times and improve the customer experience.

**Table 1.** Cloud computing vs. edge computing in the IoT architecture.

| Contents | Cloud Computing | Edge Computing |
| --- | --- | --- |
| Computing model | Centralized. | Distributed. |
| Computing power | The linear growth of computing power in the cloud cannot meet the increased demand for massive multi-source data processing at the network edge in IoT architectures. | Data processing power is enhanced by performing computation close to the endpoint of the IoT device [8]. |
| Network performance [9,10] | With massive access to IoT devices and massive amounts of data, network bandwidth and transmission speeds have reached bottlenecks. | Workloads published for the edge can reduce latency and bandwidth, network performance is optimized. |
| Real-time | Non-real-time, responsible for extensive data analysis of long-period data. | Focuses on the analysis of real-time, short-period data. |
| Availability | If the cloud center goes down, all IoT devices that rely on the cloud data center will not be available. | The edge requires that edge services can continue to operate in a disconnected or weak network state. |
| Privacy and Security | Data has a long path through the transport layer to the cloud center, which can easily lead to the loss or leakage of private user data. | Reducing transmission distances, avoiding privacy breaches and edge-side security need attention. |
| Energy consumption | Energy consumption is high and data centers consume tremendous energy [11]. | The relatively low energy consumption reduces costs. |

After several years of technology reserve and development, edge computing has obtained a lot of innovative achievements in related technologies and applications, etc. Existing articles tend to focus more on the study of edge computing [12] and the challenges [13], network performance [9,10], security [14] and scenario applications [15] of edge computing in the IoT domain. Currently, there is no unified academic standard for exploring the architecture of EC-IoT and there needs to be more relevant technologies and specific applications regarding EC-IoT solutions. As shown in Figure 1, from the application of edge computing in the IoT field, the development of IoT edge computing is comprehensively studied through the EC-IoT architecture. It is hoped to bring inspiration to the researchers and enterprises engaged in the field of EC-IoT.

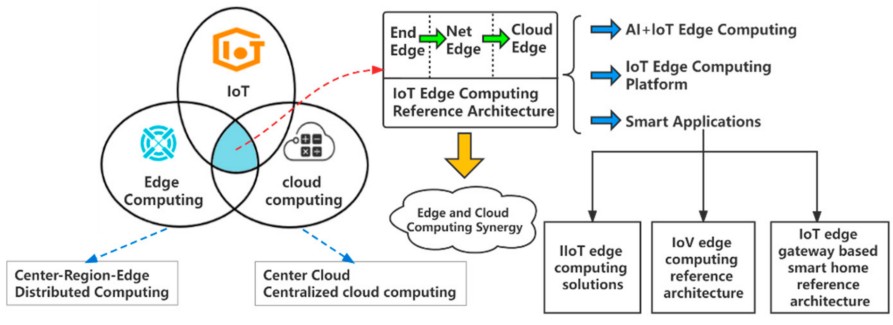

**Figure 1.** The Connection Between IoT, Edge Computing and Cloud Computing.

The specific contributions are as follows:

- Introducing edge computing into the IoT architecture. Based on the IoT architecture, a three-tier EC-IoT architecture is proposed for the end edge, network edge and cloud

edge. The advantages and shortcomings of the EC-IoT architecture are also studied and analyzed.

- The impact of artificial intelligence on EC-IoT architectures is examined, with summaries for task offloading, virtual machine (VM) migration and edge caching techniques.
- A comparative study and classification of five open-source platforms for edge computing, the NebulaStream management system and the VergeDB database is studied.
- Based on the EC-IoT reference architecture, the IIoT edge computing solution, the Internet of Vehicles (IoV) edge computing reference architecture and the edge gateway smart home reference architecture are proposed. Some of the new challenges encountered are also discussed.
- Finally, future research directions and some open challenges in IoT-Edge Computing are summarized.

The remaining sections are organized. Section 2 introduces the definition of edge computing, proposes the EC-IoT architecture and analyses the strengths and weaknesses of the EC-IoT architecture. In Section 3, the impact of AI on the edge computing architecture of the Internet of Things is deeply studied. In Section 4, the IoT edge computing platform is discussed. In Section 5, the application and challenges of the EC-IoT architecture in IIoT, Internet of vehicles and smart home are explored. In Section 6, future research directions and some open challenges of IoT edge computing are summarized. Finally, Section 7 summarizes this article.

## 2. Architecture of IoT Edge Computing

In this section, the definition of edge computing is introduced, the integration of edge computing technologies with existing IoT architectures, the EC-IoT reference architecture is proposed, the key technologies at different levels and layers of the EC-IoT reference architecture are highlighted, and a comparative analysis of the EC-IoT strengths and weaknesses is presented.

### 2.1. Definition of Edge Computing

The prototype of edge computing can be traced back to 1998 when Akamai launched the Content Delivery Network (CDN) [16], which at that time was only responsible for storage and data marginalization. After that, it experienced the birth and evolution of cloudlet [17–19], mobile edge computing [20,21], fog computing [22–24] and cloud-sea computing [25] and other related concepts until 2013, when the term edge computing was first proposed [11]. International standards organizations [26,27], enterprises [28,29], industry [30], academia [31–33], etc., also have defined edge computing. While there are differences in the current definitions of edge computing, the consensus remains that cloud computing capabilities are sunk to edge nodes with the help of edge networks. This paper combines definitions from various parties: edge computing is a new distributed computing model that integrates cloud, network, end and intelligence. Edge computing combines the computing power, storage and application resources of edge nodes with distributed cloud computing technology, thus shortening response time and reducing bandwidth requirements. Edge computing provides efficient capability support for edge applications such as Telematics, intelligent manufacturing and ultra-high-definition video broadcasting.

Edge computing mainly focuses on business scenarios such as real-time, short-period data and local decision-making. It is more suitable for integration into IoT architectures to provide efficient and secure services to many end users.

### 2.2. IoT Architecture for End-Net-Cloud Edge Computing

As shown in Figure 2, this architecture aims to introduce edge computing on any device and network path between IoT devices, gateways and cloud infrastructure. By enriching the end, network and cloud capabilities of IoT, three types of EC-IoT are formed: End edge, Network edge and Cloud edge, which provide different levels of service capabilities close

to data sources for various scenarios, thus improving the performance of IoT devices in terms of data processing and real-time performance.

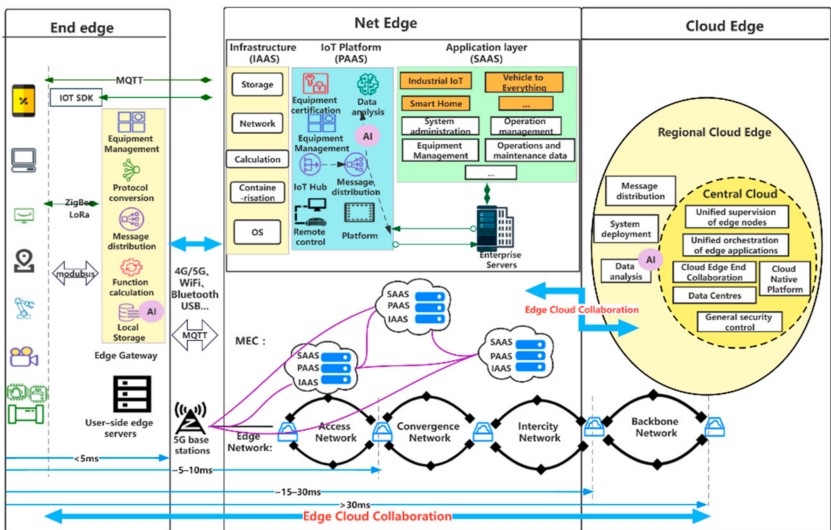

**Figure 2.** IoT Edge Computing Reference Architecture.

In the EC-IoT reference architecture, the central cloud services are gradually extended along the network nodes to build multiple small and medium-sized edge cloud servers that can provide the computing and resources required to host edge applications in the end edge, network edge and cloud edge. Each edge cloud mainly includes three core cloud center service models: Infrastructure as a Service (IaaS), Platform as a Service (PaaS) and Software as a Service (SaaS). IaaS: Provides services such as hardware, data centers, network components and storage. PaaS: Provides a platform that allows users to deploy and run new applications. SaaS: Provides software or applications to do and store users' work online [34].

End edge: The end edge consists of various sensors, controllers, gateways and other IoT devices. IoT sensor devices and controllers collect data through sensors using wireless networks and support the management and storage of computing resources. Based on control commands from the end-edge cloud server, the controller receives, processes and forwards data and can support initial analysis and filtering of edge data. The edge gateway can collect data collected by sensor devices via wired networks (such as fieldbus, industrial ethernet, industrial optical fiber, etc.) or wireless networks (such as Wi-Fi, Bluetooth, RFID, NB-IoT, LoRa, 5G, etc.) and act as an edge cloud server to provide heterogeneous computing [35]. Edge gateways can combine data from various devices to provide a broader range of judgements than sensor devices and controllers. Edge gateways also have device management capabilities. In addition, the edge gateway can transmit control streams from the network edge or cloud center to the device controller at the end edge. The end edge layer optimizes the network latency to around 5ms and uploads the filtered data to the network edge.

Net edge: The entire network transmission path and edge node layer between the end edge and the cloud edge, mainly comprising the edge network and the Multi-Access Edge Computing (MEC) edge cloud platform. The edge network provides the connectivity required for central cloud computing to sink to the edge, supporting proximity access and edge shunting. The ETSI's standard definition of MEC [27] is to provide an IT service environment and cloud computing capabilities at the edge of the mobile network. The MEC edge cloud platform manages large amounts of heterogeneous data generated at the network's edge from different types of IoT devices. It provides data processing, storage and more accurate model training capabilities [36]. The IoT Cloud Service powered by MEC integrates industry-oriented AI solutions and is a new programming model for large-scale

IoT applications. The entire network edge architecture requires not only the operational and virtualization management capabilities of an IoT edge computing platform, but also the deployment and scheduling capabilities of edge-side business application management [35]. Such an architecture enables better resource allocation and production scheduling decisions at the edge of the network, enabling optimal measures to be taken over a wider area and in a shorter time frame. Such an architecture would allow for the computation of downstream data on behalf of cloud services at the network's edge and upstream data on behalf of IoT devices [37].

Cloud edge: An extended extension of the central cloud computing service to the cloud edge, primarily for enabling the mining of massive amounts of data. This layer extends the centralized cloud capability to the regions, enabling full regional coverage. Its ability to provide robust, more proximal data processing, system deployment, message distribution, etc., at the cloud edge solves the time-consuming in the backbone network and optimizes the network latency to about 30 ms, but is still logically a central cloud service. Thanks to its proximity to the cloud center, the cloud edge enables optimal allocation of resources across enterprises, regions and even the country [35]. In addition, the central cloud can perform unified monitoring and management. The central cloud can issue all kinds of data and instructions to the end, network and cloud edge side for execution, dynamically adjusting the deployment policies and algorithms on the edge side according to the distribution of network resources. The central cloud effectively improves the utilization of computing resources while ensuring low service response latency.

The EC-IoT reference architecture proposed in this paper is essentially "connectivity + proximity computing [38]" to achieve optimal overall benefits. However, in practice, edge computing faces more complex scenarios, from the data source to the cloud center, where edge cloud capabilities such as data, storage and computation can be distributed throughout the edge node layer between the IoT device and the data center, depending on demand. Therefore, the transmission path of the edge network between the central cloud and the terminal is complex. The respective latency generated by different edge locations varies, being via the access network (latency about 5~10 ms), aggregation network, metro network (latency about 15~30 ms) to the backbone network (latency greater than 30 ms) and finally to the data center.

### 2.3. Advantages of IoT Edge Computing Reference Architecture

Compared to the traditional IoT architecture, the EC-IoT reference architecture proposed in this paper achieves all-around collaboration at the End edge, Network edge and Cloud edge, realizing a unified regulatory model with one central cloud and multiple edge clouds. The EC-IoT reference architecture will offer more significant advantages in the face of the challenges of IoT architecture applications in the 5G era.

- Faster response times: When workloads are published at the edge and require local data input, processing can be performed closer to the edge where the data is generated, effectively reducing latency and increasing responsiveness for real-time or near-real-time data analysis and processing.
- Reduced bandwidth consumption: Edge computing enables data to be stored and processed at the edge, which can simultaneously avoid the impact of large-scale traffic on the network substantially, with a significant reduction in data volume and transmission distance, reducing the bandwidth consumption of the local network.
- Intelligent: Empowering the edge with innovative capabilities, thoughtful analysis at the edge, extracting and aggregating the data needed through intelligent analysis, eliminating useless data, driving applications towards intelligence and realizing automatic feedback and smart decision-making.
- Security: Data is generated, processed and stored on the edge device, avoiding the leakage of sensitive data due to data transfer between the device and the cloud. In addition, keeping the data local to the device maintains the integrity of the data.

- Cost-effective solutions: Network bandwidth, data storage and computing power incurs certain upfront costs, and the transmission of large amounts of data over long distances leads to high-cost consumption. In contrast, edge computing performs data computing tasks locally, reducing the final cost of the IoT solution.

### 2.4. Advantages of IoT Edge Computing Reference Architecture

EC-IoT reference architecture has the characteristics of decentralized and diversified application deployment for edge computing. In this regard, the practical application of the EC-IoT reference architecture poses new problems compared to the IoT architecture.

- Successful deployment of EC-IoT reference architectures requires a robust infrastructure of edge resources, radios, base stations and terminals. The EC-IoT reference architecture allows organizations to increase their computing power faster and at a lower cost, with the attendant higher costs of infrastructure construction and operations and maintenance. At the same time, the cost of deploying applications across a multi-cloud infrastructure will be compensated by the benefits it offers.
- The EC-IoT reference architecture uses a unified supervisory model with one central cloud and multiple edge clouds. IoT edge computing nodes have limited resources and need to distribute and schedule tasks according to the type and scale of the actual tasks. The unified partitioning of complex tasks through cloud-network-edge collaboration, while considering the heterogeneity of hardware and software and the resource capacity of edge nodes, also leads to a more complex control logic for data management and query execution in EC-IoT.
- In the EC-IoT reference architecture, numerous sensors and devices generate vast amounts of data, with different third-party providers providing all the storage. The outsourcing of user data to these storage providers, whose storage devices are deployed at the edge of the network and located at many different physical addresses, increases the risk of attack [35]. At the same time, due to the open nature of its computing power, edge computing also poses security risk issues in terms of applications, data, networks, infrastructure, physical environment, and management [39].

## 3. The Impact of Artificial Intelligence on IoT Edge Computing Architectures

With the rapid development of information technology, many emerging technologies, including edge computing, have been created. The future development of the EC-IoT architecture is bound to be a convergence of developments with other emerging technologies, such as Edge Intelligence (EI) [40–42]. AI works to simulate intelligent human behavior in devices and machines by learning from data [43]. The convergence of AI and IoT has contributed significantly to the rapid development of Artificial Intelligence IoT (AIoT) systems. Driven by this trend, there is an urgent need to push the AI frontier to the network edge to unlock the full potential of big data in the EC-IoT reference architecture, resulting in a new research area—EI. EI is a new paradigm for EC-IoT and AI to empower each other, focusing on enabling intelligent applications and protecting user privacy in edge environments with the help of edge computing [44]. This section examines the application of EI to EC-IoT architectures in three sections: Task offloading, VM migration and edge caching.

### 3.1. Task Offloading

The practical application of the EC-IoT architecture often generates a large amount of AI computing power demand. When the computing power of the mobile terminal itself cannot be satisfied, the computing tasks will be fully or partially offloaded to a nearby edge cloud server with high computing and storage resources, and the results will be returned to the terminal when the computing is completed. Zaman et al. [45] mentioned that AI and machine learning, whether deep models or lightweight models, are designed to offload decision optimization. Offloading computing tasks in EC-IoT architectures is a crucial research question. That is, whether computational tasks should be executed locally or fully or partially offloaded to edge cloud servers or the cloud, and how network, computational

and storage resources should be allocated at the edge. Different task offloading schemes significantly impact task completion latency and mobile device energy consumption [46].

Xiao et al. [47] proposed an innovative solution with Edge Intelligence for computational tasks offloading for dependent IoT applications (CODIA) with better robustness and efficiency in terms of convergence, latency and energy consumption. Chen et al. [48] proposed a game theoretic approach for the computation offloading decision making problem among multiple mobile device users for mobile-edge cloud computing and also designed a distributed computation offloading algorithm with excellent computation offloading performance. Ali et al. [49] proposed a novel energy-efficient deep learning based offloading scheme (EEDOS) to train a deep learning based smart decision-making algorithm on achieving high accuracy for optimal decision making for the offloading problem in MEC.

Current MEC offload solutions mostly separate compute offload from user mobility. In particular, in MEC, the unknown location, speed and direction of the user on the mobile device will affect the EC-IoT architecture application system latency. Zaman et al. [50] proposed a framework termed COME-UP Computation Offloading in mobile edge computing with Long-Short Term Memory (LSTM) based user direction prediction. The framework effectively reduces delays and energy consumption and improves resource efficiency.

### 3.2. Virtual Machine Migration

In EC-IoT architecture applications, where fixed or mobile devices have limited computing power and energy, MEC pushes these services from local IoT devices to the network edge by deploying cloud servers near user devices at the network edge. Mobile devices can collaborate with the network edge to achieve a higher quality of service and low latency through task offloading.

Many applications and services running on mobile devices, given the mobility of users and the limited edge resources available to serve them, may require users to communicate with cloud servers over multiple hops, which can severely impact communication latency and quality of service (QoS) between users [51]. To reduce the negative impact of this QoS degradation, user mobility becomes an essential driver for the real-time migration of VMs for EC-IoT application scenarios. When a user moves out of the coverage area of the current edge cloud data center, the VM is migrated to the edge cloud data center where the user currently is [52]. The location, speed and direction of the user, the compute migration area and the location where compute migration can be performed will all affect the VM migration policy [53]. Therefore, there are challenges in designing the right migration solution.

Shahryari et al. [51] proposed a novel Cost-aware VM placement and migration (CoPaM) framework for mobile services in a network of cloudlets. Osanaiye [54] proposes a conceptual smart pre-copy live migration approach, which is presented for VM migration that will minimize both the downtime and the migration time to guarantee resources. Mangalampalli et al. [55] proposed a novel deep learning network WBATimeNet, which uses Multivariate Time Series data of Memory, CPU and Disk to predict which VM should be live migrated.

### 3.3. Edge Caching

Edge caching is a crucial technology needed to enable cloud-side intelligence collaboration at the edge of the network in the EC-IoT architecture. In EC-IoT network architectures, the same content in the network is often requested multiple times by smart end devices in the same area. Leveraging edge caching can reduce heavy traffic loads and end-to-end latency in radio access networks (RANs) and reduce duplicate transmissions of the same content to support many critical IoT services and applications [56]. Caching in the MEC increases network capacity by making content available locally, saving network bandwidth [57].

Yasir et al. [57] proposed a user preference-aware content caching (CoPUP) scheme for the MEC environment that can cache the most popular content, reduce response time and enhance the cache hit ratio. Gupta et al. [58] proposed an enhanced ICN-IoT content

caching strategy by enabling AI-based collaborative filtering within the edge cloud to support heterogeneous IoT architecture. Zhang et al. [59] proposed a deep reinforcement learning based cooperative caching approach for IoT edge caching. The approach allows the distributed edge servers to learn to cooperate with each other.

EI supports the EC-IoT reference architecture for complex and dynamic industrial tasks in wireless environments. Liu et al. [60] used multi-intelligent Deep Reinforcement Learning (DRL) to achieve collaborative resource allocation at the end edge. Compared to traditional DRL, the proposed approach can better adapt to the edge's computational power, the data's size, the required computational resources and the number of industrial devices. Foukalas et al. [61] proposed a federated active migration learning (FATL) model based on EI, which can address challenges in industrial IoT responsiveness and security through training and testing. Hayyolalam et al. [62] proposed a novel model of intelligent healthcare to improve the utilization of EI technology in intelligent healthcare systems.

In addition, developing the EC-IoT architecture is equally relevant to developing technologies such as cloud computing [63], big data [64,65], 5G [66,67], arithmetic networks [68], cloud-native [69] and blockchain [70,71]. The convergence of EC-IoT architecture with technologies such as AI provides technologies and methods for EC-IoT, in addition to the advantages of edge computing, such as low latency and reduced bandwidth, where data generated at the edge of the network can be unlocked to its full potential and scalability. At the same time, EC-IoT also provides scenarios and platforms for technologies such as AI, extending the range of applications. The introduction of technologies such as AI into EC-IoT architectures is set to generate a wealth of innovative research findings that will provide practical solutions to the complex challenges faced by enterprises in EC-IoT architectures.

## 4. IoT Edge Computing Platforms

As the EC-IoT architecture grows in application size, complexity and overall demand, the functional requirements for the EC-IoT system architecture vary significantly from application scenario to application scenario. Therefore, there is an urgent need to form a standard EC-IoT platform with unified operation and maintenance, unified control and unified delivery to provide secure and reliable computing and management services for the collaboration of the end edge, network edge and cloud edge in the EC-IoT architecture.

### 4.1. Open Source Platform for Edge Computing

The Edge Computing platform is now one of the main focuses of researchers as an integrated solution that can simultaneously address the issues of data heterogeneity, manageability of computing resources and application complexity. Table 2 compares five open-source edge computing platforms: EdgeX Foundry [72], EdgeGallery [73], Akraino Edge Stack [74], KubeEdge [75] and OpenYurtp [76].

In line with the service objectives, EdgeX Foundry focuses on the end-edge of the EC-IoT architecture. It works to solve problems in the development and deployment of IoT applications. EdgeGallery and Akraino Edge Stack are focused on providing MEC edge cloud or edge cloud services to optimize or rebuild the infrastructure at the network's edge to provide cloud-like services at the edge of the network [77]. The cloud-edge collaborative edge computing platforms, represented by KubeEdge and OpenYurt, are designed to migrate cloud solutions to IoT devices with the concept of "cloud-edge convergence".

In addition, other related companies and organizations have launched a range of edge computing platforms. These include the Open Networking Foundation's (ONF) CORD platform, the OpenStack Foundation's StrlingX, Amazon's AWS IoT Greengrass, Microsoft's Azure IoT Edge, IBM's Apache Edgent, donated to the Apache Software Foundation, the University of Wisconsin-Madison WiNGS Lab developed ParaDrop [78,79], AliCloud's Link IoT Edge, Baidu Cloud's Baetyl, Tencent's SuperEdge, EMQ's EMQ X Kuiper, etc. As shown in Table 3, based on the EC-IoT architecture design goals and requirements, the more active, innovative and popular open-source platforms for edge computing in EC-IoT can be divided into three categories: IoT user side, edge service side and cloud-side collaboration.

**Table 2.** Comparative study of edge computing platforms.

| Projects | EdgeX Foundry | EdgeGallery | Akraino Edge Stack | KubeEdge | OpenYurt |
|---|---|---|---|---|---|
| Vendors | Linux Foundation | Huawei and others | Linux Foundation | Huawei | Alibaba Cloud |
| Open source or not | Yes | Yes | Yes | Yes | Yes |
| CNCF Project | No | No | No | Yes | Yes |
| LF Edge Project | Yes | Yes | Yes | No | No |
| Cloud Edge Collaboration | No | Support | Support | Support | Support |
| Cloud Native K8s Eco-Compatible | No | No | No | Partially compatible | Full compatibility |
| Edge Autonomy | Stable operation with intermittent connections | Support | NO | Support | Support |
| Deployment Complexity | Complex | Simple | Complex | Simple | Simple |
| Containerized Orchestration | NO | NO | NO | Support | Support |
| Service Objectives | IoT End Edge | 5G MEC Edge Cloud | Edge Cloud | Cloud Edge All-in-One | Cloud Edge All-in-One |
| Application Scenarios | Provides end-edge solutions. Mainly in industrial IoT scenarios. | 5G MEC edge cloud solutions. Smart manufacturing and other application scenarios. | Total solutions for edge infrastructure. Application scenarios such as smart cities. | Side-end cloud collaboration solutions. Smart factories and other industries. | Cloud edge collaboration solutions. Smart logistics and other industries. |

**Table 3.** Edge computing platform classification table.

| Classification | Enterprise Institutions | Open Source Platform for Edge Computing |
|---|---|---|
| IoT user-side edge computing open-source platform | Linux Foundation<br>Apache Software Foundation<br>WINGS Lab, University of Wisconsin-Madison<br>EMQ | EdgeX Foundry<br>Apache Edgent<br><br>ParaDrop<br><br>EMQ X Kuiper |
| Open-source platform for edge computing on the edge service side | Open Network Foundation ONF<br>Linux Foundation<br>OpenStack Foundation Hosting<br>Huawei/CAICT | CORD<br>Akraino Edge Stack<br>StrlingX<br>EdgeGallery |
| Cloud Edge Collaborative Edge Computing Open Source Platform | Microsoft<br>Huawei<br>Alibaba Cloud<br>Tencent<br>Baidu<br>Amazon | Azure IoT edge<br>KubeEdge<br>OpenYurt, Link IoT Edge<br>SuperEdge<br>Baetyl<br>AWS IoT Greengrass |

By deploying on the edge side of the EC-IoT architecture, the edge computing platform is well-positioned to solve the problems of massive data processing and massive device terminal connectivity. In addition, the edge computing platform enables end-net-cloud edge coordination, while providing hardware and software support for applications based on the EC-IoT architecture. However, with the rapid development of edge computing platforms, there are still many problems with their application in the EC-IoT architecture.

- The selection and construction of scenarios for deploying edge computing platforms in EC-IoT architectures must be based on actual business needs, the type of solutions generated, the skills required to organize the solutions generated and the long-term maintenance of these solutions [80].

- As EC-IoT architecture application scenarios continue to grow, applications such as smart homes, intelligent transportation and smart city are also receiving more and more attention. Edge computing platforms will face fundamental challenges in systematically supporting the functional requirements of IoT edge application scenarios, achieving more simplified deployment and rapid scaling of edge cloud services, and improving the reliability of standard operating systems.

- The Cloud Native Computing Foundation (CNCF) defines cloud native as enabling organizations to build and run elastic and scalable applications in new dynamic environments such as public, private and hybrid clouds [81]. Cloud-native technologies and concepts, including Kubernetes (K8s), containers and microservices, emphasize loosely coupled architectures and the ability to scale quickly and conveniently, aiming to achieve a consistent cloud computing experience across different infrastructures through uniform standards. For EC-IoT application scenarios, cloud-native technology can provide integrated application distribution and collaborative management for the cloud-side end, solving the problems of edge-side large-scale application delivery, operation and maintenance, and control. As a result, some vendors have launched a series of edge computing platforms based on K8s, such as Huawei's KubeEdge and Alibaba Cloud's OpenYurt. With the increasing demand for cloud-native development for applications related to EC-IoT architecture, the need to accelerate the construction of cloud-native infrastructure platforms has become more and more urgent.

- Due to the complexity of heterogeneous resource support, diverse communication methods and scattered distribution locations of edge-end devices, edge computing platforms managing EC-IoT architecture edge devices often need to address additional issues, such as data storage complexity. In response, several examples of edge computing platform collaboration have been proposed. In 2021, Alibaba Cloud and VMware proposed an integrated cloud-edge-end platform based on OpenYurt and EdgeX Foundry, which further realizes the collaboration of "cloud, edge and end" and creates an integrated and collaborative IT architecture of cloud-edge-end. This paper argues that: The collaborative development of multiple edge platforms will not only help migrate cloud solutions to IoT devices but also further drive the implementation of cloud-native projects in the EC-IoT space while guiding more enterprises and developers on experiences they can learn from. In this regard, it will also be a significant challenge to collaborate among edge computing platforms in the EC-IoT architecture in the future to improve efficiency and maximize resource utilization.

*4.2. IoT-Related Platforms*

In addition to the Appeal Edge Computing platform, we have recently discovered several exciting and new IoT-related platforms.

- NebulaStream

NebulaStream [82,83] is a general purpose, end-to-end data management system for the IoT. It provides an out-of-the box experience with rich data processing functionalities and a high ease-of-use. NebulaStream aims to provide unique aggregation capabilities, innovates and integrates various technologies, including cloud, fog and sensor networks, to create a unified sensor-fog-cloud environment and facilitate the development of foreseeable IoT applications. Furthermore, the system follows the design principle of maximized sharing [84]. NebulaStream addresses the challenges of heterogeneity, unreliability and scalability of EC-IoT by unifying the end, network and cloud edge infrastructure to enable real-time processing of data and efficient utilization of underlying heterogeneous and geo-distributed resources.

- VergeDB

VergeDB [85] is a database for adaptive and task-aware compression of IoT data that supports complex analytical tasks and machine learning as first-class operations. VergeDB serves as either a lightweight storage engine that compresses the data based on downstream

tasks or as an edge-based database that manages both compression and in-situ analytics on raw and compressed data. In EC-IoT architectures, an edge-based database with adaptive compression and support for complex analytics is required to minimize the amount of data transferred to the cloud while supporting in-situ operations. In this regard, VergeDB can be the first step in realizing this vision.

## 5. Applications and Challenges of IoT Edge Computing Architectures

Market research firms and markets forecast the global edge computing market to reach US $87.3 billion by 2026, from US $36.5 billion in 2021, at a CAGR of 19.0% during the forecast period [86]. This covers the entire EC-IoT industry, including hardware, software, services, edge platforms and vertical markets. According to International Data Corporation (IDC), the total global IoT data will be 16ZB in 2020, of which 45% of IoT-generated data will be processed at the network edge. The Gartner study also predicts that the proportion of enterprise-generated data processed outside of centralized data centers will jump from 10 percent in 2018 to 75 percent by 2025 [87].

IoT supports many vital applications, including intelligent traffic management, safety-aware autonomous driving, electricity savings using smart grids, innovative industrialization and intelligent home solution [88] For latency-sensitive IoT applications, long processing times are intolerable [89]. Using EC-IoT architecture has led to significant performance improvements in real-time, bandwidth consumption, cost and privacy and security of related applications. However, due to the heterogeneity, complexity, timeliness and relevance of the IoT domain, there are still many technical areas for improvement in developing edge computing in IoT application scenarios. shown in Figure 3, this section will look at the EC-IoT reference architecture and discuss the solutions and challenges of the architecture in the IIoT domain in the Telematics and Smart Home application scenarios.

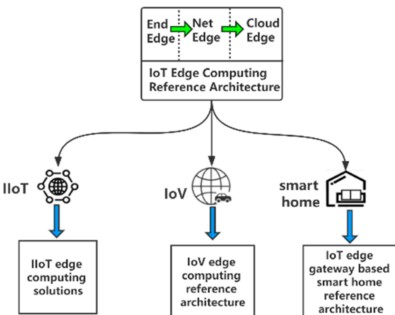

**Figure 3.** Application scenarios for the EC-IoT reference architecture.

### 5.1. Applications and Challenges of IoT Edge Computing in Industrial IoT

IIoT is the application of IoT in the industrial sector, deep integration of IoT and traditional industries. In recent years, IIoT has entered a new phase, benefiting from the addition of 5G, edge computing and AI technologies. The combination and collision of mechanistic and data models have provided powerful data support and new engine power for resolving uncertainty in complex systems and business decisions. With the full development of IIoT, EC-IoT racking is already a critical step in building IIoT systems. The deployment of edge computing will drastically reduce the cost of IIoT production brought about by the network, cloud data center computing and storage, further driving the realization of intelligent manufacturing and bringing enormous opportunities for IIoT development. In response, as shown in Figure 4, this paper proposes an industrial EC-IoT solution based on the EC-IoT reference architecture.

In industrial EC-IoT solutions, edge nodes assume edge arithmetic power for AI inference and enable edge-side data uploading to edge cloud platforms through cloud-side collaboration capabilities. The edge cloud platform can support various device access through multiple protocols downwards, accept centralized management of edge nodes from the central cloud upwards, and perform real-time edge data processing, intelligent

computing and decision-making on business data according to the model parameters issued by the central cloud. At the same time, the edge cloud platform can still work typically when communication with the central cloud is interrupted, achieving edge autonomy. Industrial EC-IoT solutions enable systems to monitor, collect, exchange and analyze data and deliver high-value decisions unprecedentedly, meeting the industry's critical needs for real-time business, data optimization, application intelligence, security and privacy protection.

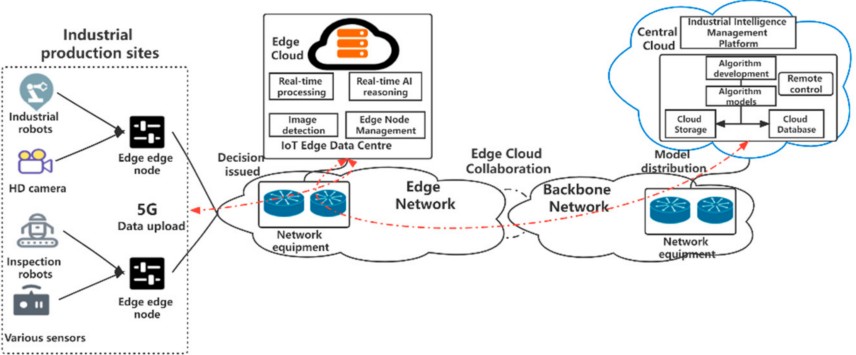

**Figure 4.** Diagram of an industrial EC-IoT solution.

Gartner states that 30% of industrial control systems are expected to have analytics and AI edge reasoning capabilities by 2025, compared to less than 5% in 2021 [90]. In response, the IIoT scenario EC-IoT will see rapid growth. Chen et al. [91] proposed an edge-computing architecture for IoT-based manufacturing based on the advantages of real-time processing and the autonomy of edge computing. The architecture has significant advantages in terms of service agility and bandwidth optimization, which help to improve the quality of service in industrial manufacturing. Qin et al. [92] investigated the integration mechanism and application methods of the core technologies of industrial robots and the critical technologies of edge computing using case studies and comparison methods. Xu et al. [93] addressed the problem that edge nodes are usually light in weight and usually low in computational power by proposing the use of hybrid cloud edge computing or multi-tier edge computing hierarchies to improve the stability and latency of computing platforms.

The optimal allocation and management of resources in a complex IIoT network environment will be a significant challenge for IIoT development. The edge layer will become increasingly blurred, and edge devices will become increasingly intelligent and diverse. However, while the intelligence and blurring of the industrial edge layer bring excellent efficiency gains, it also poses significant security challenges.

- In the IIoT space, the security quality of edge devices from various manufacturers will be difficult to guarantee as there is still a need for accepted standard specifications for edge computing.
- Edge devices need to be exposed to the internet to interface with cloud platforms and are bound to encounter various security issues when dealing with data from various industrial protocols.
- The addition of edge computing capabilities to some industrial devices or terminals will break the constraints of the original centralized security management. There are bound to be security loopholes in smart devices in this model. if these loopholes are exploited, they may cause serious production accidents.
- The rapid growth of edge devices is accompanied by increasing energy consumption, resulting in an increasingly challenging energy situation for IIoT systems. For example, the cost of downtime due to faults and unpredictable power disturbances is expensive in the case of smart grids. The main problem with smart grids is the need to collect large amounts of data from IoT devices, and processing the data is a challenge. EC-IoT makes it possible to analyze data in real time and to keep edge services running even

in the event of a disconnection brought on by a fault, so that problems can be avoided in advance or the cause of the problem can be determined more quickly. All this with a high degree of security. A key challenge for EC-IoT systems is to reduce costs while still fulfilling the task of offloading. Albataineh et al. [94] proposed a hybrid solution by using the Cloud and Edge Computing to process the data. Aiming at the problem of service offload scheduling in edge computing. Xing et al. [95] proposed a delay optimized task offload algorithm based on task priority classification. This algorithm can effectively improve the overall system revenue and reduce user task delay.

With the rapid development of IoT edge devices, this paper calls on the relevant authorities in each country to join major edge device manufacturers and security vendors together as soon as possible to initiate the development of relevant security standards and specifications to ensure that business needs are met while constraining security and interconnection to promote the rapid and secure development of IIoT.

### 5.2. Application and Challenges of IoT Edge Computing in Internet of Vehicles

The Internet of Vehicles (IoV) is an important research area in IoT smart mobility, which is based on sharing information in the form of Vehicle to Infrastructure (V2I), Vehicle to Vehicle (V2V), Vehicle to Pedestrian (V2P), Vehicle to Self (V2S) and Vehicle to Roadside Unit (V2R) [96]. IoV requires dynamic data generated with a latency of 10ms or less to be processed in real-time. However, network communication can become unstable when the vehicle moves, resulting in high latency in data upload. From a business perspective, the massive data transmission and storage cost is also a big issue facing IoV [97]. In this regard, it is shown in Figure 5. This paper proposes the IoV edge computing architecture based on the EC-IoT reference architecture. The architecture includes three layers: Vehicle/person/road edge side, network edge and cloud center.

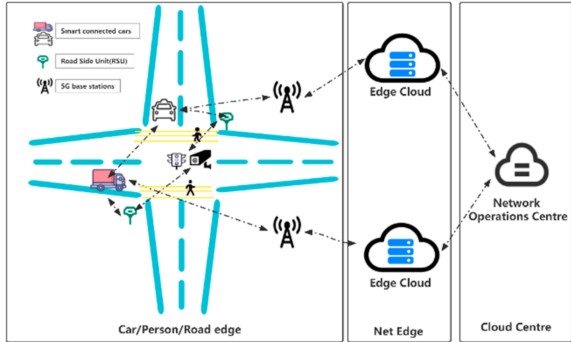

**Figure 5.** Internet of Vehicles Edge Computing Architecture.

Vehicle/people/road edge side: While the vehicle is in motion, the roadside unit will acquire roadside personnel, vehicles, temperature and humidity and other road condition data through sensing devices such as cameras, street light controllers, weather sensors and radar. The vehicle will aggregate the data from the vehicle itself and the roadside unit and process it in real-time at the vehicle's edge node while transmitting it to the edge cloud for remote real-time processing, analysis and decision-making through a 5G base station.

Net edge: The net edge mainly provides access and management capabilities for vehicle-side devices, remote real-time processing, analysis and decision-making. Vehicle and roadside unit side computing tasks can be offloaded to the nearby edge cloud server to perform AI real-time analysis of data from vehicles and provide application services. It mainly includes services such as dynamic speed limit warning, congestion analysis, dangerous driving alert, violation detection, vehicle convergence warning and pedestrian detection.

Cloud Center: To meet the operational needs of global traffic information among vehicles, road measurement units and application platforms, the vehicle-road collaboration architecture hands over complex data processing, analysis and control strategies to the

central cloud through cloud-edge collaboration. The vehicle-road collaboration architecture realizes the management, analysis and decision-making functions of multiple edge clouds, edge nodes and vehicles in non-real-time classes. Cloud-edge collaboration provides dynamic real-time information interaction of vehicle–vehicle, vehicle–road and human–vehicle from multiple aspects for vehicle safety control and collaborative road management.

For IoV applications with high mobility of user devices, the location, speed and direction of the user's vehicle also have a great impact on the latency of the IoV system. Hoang et al. [98] proposed an analytical model to calculate the offloading decision considering the random movement of vehicles and the possible handover problem in the offloading process. In order to cope with the dynamic changes of the offloading environment for computing resource-intensive and separable tasks in IoV, Cao et al. [99] proposed a distributed offloading strategy that multiple collaborative nodes had serial offloading mode and parallel computing mode in the vehicle to everything (V2X) scenario was proposed. This strategy can effectively reduce the system delay of computing tasks.

A new report from Research And Markets [100] shows that edge computing infrastructure and services to support self-driving cars are expected to reach $42.8 billion by 2027. In August 2017, Ericsson, Toyota and others formed the Automotive Edge Computing Consortium (AECC) to drive the development of smart cars. Academically, Zhang et al. [101] built an open vehicle data analysis platform OpenVDAP based on edge computing, which provides a full wharf of vehicle data computing services such as a vehicle computing platform, operating system and function library. Tang et al. [102] developed LoPECS, the first complete edge computing system for producing self-driving cars, which leverages the runtime layer of heterogeneous computing resources of low-power edge computing systems to meet the real-time requirements of self-driving applications. Ibn-Khedher et al. [103] proposed an end-to-end architecture for edge-assisted autonomous driving that allows the rationing of computationally intensive autonomous driving services to shared resources on edge servers, improving the performance level of autonomous vehicles.

The challenges of EC-IoT architecture in IoV applications are as follows:

- EC-IoT applications in IoV involve rich scenarios such as machine vision, big data processing, acoustic detection, vehicle tracking, etc., which require different computing, storage and network resources to provide support. The massive fragmented EC-IoT device environment will significantly limit the implementation of IoV application.
- Since edge computing systems in IoV are mobile, they have stringent energy consumption constraints [104]. Providing sufficient computing power, redundancy and security with reasonable energy consumption to ensure the safety of self-driving vehicles is one of the challenges in designing IoV edge computing systems.
- Limited by the computational capacity of edge nodes and the importance of edge node latency on data processing speed, edge node computational resource scheduling and selection are also issues to be considered.

This paper argues for more excellent investment in research into optimal edge node selection schemes and multi-node distributed collaborative computing methods in transportation systems. According to the real-time requirements of applications, the optimal deployment of resources should be combined with computing resources, network delay, bandwidth and energy consumption. At the same time, intelligent transportation and autonomous driving development should be promoted by the interactive fusion of information, collaborative decision-making and control among various traffic participants. However, as there are currently no fully functional self-driving vehicles, further requirements and applications in this area will undoubtedly emerge in the next decade [105]. Exploring solutions to these challenges will be essential to making the EC-IoT architecture available to serve the more extensive future transportation system.

### 5.3. Applications and Challenges of IoT Edge Computing in Smart Home

The smart home is a crucial application scenario of the IoT architecture. Additionally, with the emergence of the EI concept, coupled with the improvement of chip performance,

edge computing has become more and more powerful while also beginning to carry more business logic. In response, more intelligent devices have emerged in the innovative home environment, extending the cloud computing capability from the intelligent home. For example, suppose an edge gateway is introduced in a smart home, on the one hand. In that case, it can realize the control of smart home devices through edge computing. The edge gateway can make decisions faster based on the received information and control home devices to execute corresponding actions. On the other hand, in the interaction scenario of different products in smart homes, edge computing will act as a central control system to realize the interconnection and scene control between devices and other needs through the collaboration of cloud computing and edge computing. As shown in Figure 6, this paper proposes an edge gateway smart home reference architecture for the end edge of EC-IoT architecture. The architecture consists of three layers: Sensing, edge gateway and platform.

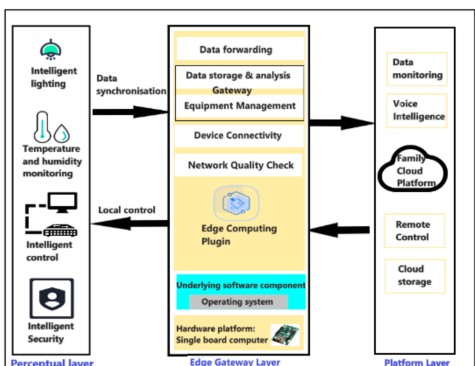

**Figure 6.** IoT Edge Gateway Smart Home Reference Architecture.

Perception layer: It mainly realizes the functions of collecting sensor data of home environment parameters and controlling smart home devices. It is the interface for nodes to access the gateway and is compatible with various communication protocols, such as MQTT, to realize the interaction of different IoT devices.

Edge gateway layer: Through the built-in edge computing plug-in to achieve cloud-edge-end collaboration, the main functions are divided into device connection, device and data management and application services. The devices are connected through the edge gateway for smart home devices in the device layer. The edge computing services within the gateway can directly process the state information of IoT home devices and collected environmental data, store the information locally and manage the real-time data. At the same time, based on the operation status of the devices, the data is used to analyze the user needs and provide suitable decision solutions.

Platform layer: It mainly collaborates with application services for data and enables human interaction with the smart home system. Users can also manage devices and monitor the gateway's status in real time through the platform layer. In addition, users can access and control smart home products remotely through the mobile phone terminal APP.

Accessing smart device status and collected environmental data based on edge gateways and the like while processing and analyzing it locally will help meet users' needs for a more comfortable, safe and convenient home living experience. According to iiMedia Research [106], the global home furnishing market will be $496.5 billion in 2021, with a year-over-year growth rate of 9.7%. Growth is expected to remain at around 3% over the next four years. There is a growing trend to adopt edge computing to help smart homes improve the comfort of residents' lives [5]. Zhou et al. [107] proposed an intelligent home electricity demand forecasting system based on EC-IoT technology, which uses a smart home gateway to store heterogeneous data in a central repository for processing and analysis, which can provide a better quality of service and scalability with limited computing resources. Li et al. [108] constructed a corresponding device management mechanism based

on a gateway near the device side as an edge server to provide an effective solution to brilliant home device diversity, heterogeneity and efficiency problems.

Many challenges remain for the adoption of EC-IoT smart homes:

- Compared with the traditional cloud center, the EC-IoT scenario lacks effective encryption or desensitization measures. Once it is hacked, its stored information of household personnel and personal privacy information will be leaked. Meanwhile, numerous insecure communication protocols (e.g., ZigBee, Bluetooth, etc.) between sensors and edge nodes lack encryption and authentication measures and are easy to be eavesdropped on and tampered with.
- The deep combination of EC-IoT and artificial intelligence, that is, the realization of intelligent home edge intelligence, from comprehensive voice control to spaced physical control and visual control, to the final realization of continuous optimization of intelligent models, active learning of the user's habits for automatic adjustment, to better provide intelligent services to users.
- In smart home systems, the rising energy costs of smart appliances such as electricity and natural gas have become a key challenge. In this regard, EC-IoT can contribute to improving the energy management efficiency of smart homes by combining it with energy management strategies. For example, Xia [109] proposed an edge-based energy management framework in the smart home scenario. At the same time, an optimal scheduling strategy is proposed to schedule the operation time of each appliance for achieving minimum electricity cost.

At present, the intelligent home systems of major innovative home manufacturers also have specific edge computing capabilities. However, overall, there is still a strong dependence on innovative home systems on cloud computing. The key people in this article's subject group will continue to explore the potential of EC-IoT in the smart home. Smart home companies fumbling around in the EC-IoT space still have a long way to go. Relevant enterprises and academics also need to actively layout to accelerate the implementation of EC-IoT in smart home scenarios.

## 6. Open Issues and Future Directions

As discussed in this paper, bringing edge computing into the IoT architecture brings many benefits. This section will discuss the main challenges of EC-IoT and the future trends.

The growing IoT market, including IIoT, Smart Grid, Smart City, Telematics, Smart Home Appliance and Smartphones, will increase the global EC-IoT market. Edge computing technologies will also dominate more in the IoT space in the future.

From the perspective of the challenges faced:

- Edge hardware: Considering the distributed deployment nature of edge computing, edge nodes may be located in various complex environmental locations. Due to the differences in deployment environments and task requirements, the hardware equipment of edge computing nodes must be comprehensively considered in the development of integration, energy consumption, hardware acceleration, robustness, security and protocol specification.
- Edge intelligence: While there have been some academic results on EI research, it is difficult to quickly complete a large number of computations on edge devices due to the weak computing power of edge devices. Moreover, the model of EI is usually complex and requires more computing resources to complete the training and inference of the model.
- Mobility issues: User mobility may lead to reduced quality of service or service disruption, especially for applications with high mobility. Further research is needed to more effectively trade off network latency against optimizing offloading decisions or migration costs to improve the quality of service.
- Edge and 5G: As 5G technology advances, the data at IoT terminals will increase. IoT edge computing will face more new application scenarios and communication demands. The EC-IoT architecture will require additional computing and forwarding

capabilities at the lower network nodes and improved management capabilities at the edge nodes of cloud services. These new demands will inevitably lead to changes in network architectures, the need for continuous improvement in edge computing capabilities and the inevitable further development of IoT edge computing.

From the future trend:

- Along with the synergistic development of edge computing and IoT, data processing power will accelerate the proliferation from the cloud to the edge, network and end edge. At the same time, computing power at the edge, network and end will continue to grow. For EC-IoT architecture, computing resources will be ubiquitous in the future.
- In IoT development, edge computing, cloud computing and hardware devices must collaborate, with cloud computing taking care of global tasks such as task scheduling and edge computing focusing on aspects such as field, real-time and security. EC-IoT architecture realizes all-round collaboration at the end edge, network edge and cloud edge. The EC-IoT architecture with unified supervision and standards for building one central and multiple edge clouds will become one of the leading development trends.

With the advent of the Internet of Everything, EC-IoT will deepen the integration and development with other emerging technologies, such as artificial intelligence and give full play to their respective advantages. It will optimise the allocation of resources from 3 aspects: end-net-cloud, to achieve improvements in system performance, user experience and cost.

In the future, the subject matter people in this paper will continue to explore the critical technologies of the EC-IoT architecture cloud, network and end edge and try to find some of the latest ideas in MEC task offloading and edge caching. The subject matter team of this paper will also try to mine the relevance of EC-IoT architecture with technologies such as 5G, blockchain and cloud-native. In addition, the solutions of EC-IoT architecture for different intelligent application scenarios will be actively explored to enrich the EC-IoT reference architecture system further.

### 7. Conclusions

An EC-IoT reference architecture was proposed based on the IoT architecture and edge computing technology. It performed a comprehensive study on EC-IoT reference architecture regarding integration with AI, edge computing platforms and application scenarios and challenges. Constructing a multi-tier EC-IoT architecture with one central cloud and multiple edge clouds with unified regulation and standards will have a very promising application, which is concluded in this paper. It is hoped to encourage academics to discuss and study EC-IoT applications and promote the community's attention and investment in EC-IoT architecture. The smart grid, smart manufacturing, smart medical, smart transportation, smart home and other industries have all demonstrated significant promise for EC-IoT. As the technology matures and new applications emerge, EC-IoT will face an increasingly large market and edge computing will become an immediate need. IoT-related companies should continue to see the new impact and more significant development of edge computing and actively construct the EC-IoT technology architecture industrial ecology.

**Author Contributions:** Conceptualization, Y.Z. and H.Y.; Methodology, Y.Z., H.Y. and M.M.; Software, Y.Z. and H.Y.; Validation, Y.Z., W.Z. and H.Y.; Formal Analysis, Y.Z., H.Y. and W.Z.; Investigation, Y.Z. and H.Y.; Resources, Y.Z., H.Y. and W.Z.; Data Curation, Y.Z. and H.Y.; Writing—Original Draft Preparation, Y.Z. and H.Y.; Writing—Review & Editing, Y.Z., H.Y. and M.M.; Visualization, Y.Z. and H.Y.; Supervision, W.Z. and Y.Z.; Project Administration, Y.Z., W.Z. and M.M.; Funding Acquisition, Y.Z. and M.M. All authors have read and agreed to the published version of the manuscript.

**Funding:** This research was funded by National Defense Basic Research Plan (grant no. JCKYS2020DC202), Natural Science Foundation of Hebei Province (grant no. F2022208002), Science and Technology Project of Hebei Education Department (Key program) (grant no. ZD2021048).

**Data Availability Statement:** Not applicable.

**Conflicts of Interest:** The authors declare no conflict of interest.

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
