# Peer review of "Application and Research of IoT Architecture for End-Net-Cloud Edge Computing"

_electronics, doi:10.3390/electronics12010001_

Round 1

Reviewer 1 Report

This vision and overview paper proposes an EC-IoT reference architecture based on architecture at the end edge, network edge, and cloud edge. It provides a relatively comprehensive comparison and evaluation of the related literature, including Edge computing and AI fields. The paper also provides a detailed framework design based on the EC-IoT reference architecture.

There are the following strengths:

  1. The paper handles a valid and vital problem for the current IoT systems. As the edge node becomes more powerful with advanced hardware, more tasks can be executed on the edge nodes, which brings more opportunities and open questions for IoT systems.
  2. The paper did extensive work on the related literature review and summarized nicely in the paper.
  3. The discussion of EC-IoT reference architecture in IIoT, Smart Grid, and Smart City scenarios raises many open questions for the community to work on.

While there are some weaknesses:

  1. The presentation of the paper needs more work. 
    1. The title of the paper is confusing to me. I guess "EC-IoT" is a superset/combination of the three-tier "End-Net-Cloud Edge Computing". These two counterparts could be redundant in the title.
    2. Please add line borders in table 2 for a better representation. The content here is a little bit redundant with table 1. The same issue existed in the corresponding text as well.
    3. Line 109-122: The cloud service models are not dedicated to the net edge. It can be applied to any tier of the model actually. So it is better to move the cloud service models into the general background discussion for section 2.2. Refactor is needed for this section.
    4. The phrase "shortening/reducing response time" is mentioned as one of the benefits of edge computing, but this statement is valid when assuming the query/workload is issued on edge and only requires the local data as input. For some large analytics workloads requiring data from multiple sources, centralized cloud computing is better regarding response time.
  2. The paper mentioned many benefits of EC-IoT, but any discussion on the drawbacks is necessary and helpful for the reader, such as the higher cost of the infrastructure and maintenance, more complicated control logic for data management (data replication, partitions), and query execution.
  3. The paper missed some interesting recent literature, such as NebulaStream and VergeDB, which provide a good classification of the challenges and workable solutions. And more system/projects form the reference can be found and included as related work for this paper.

Minor suggestion on the paper structure:

  1. Sections 2 and 3 would be better if there was a small introduction paragraph before each subsection.

Reviewer 2 Report

The article assigned to me for the review is titled “loT Architecture Study on EC-IoT End-Net-Cloud Edge Computing”. The topic is important and nicely explained, however the manuscript is lacking the following points.

-  The title is not in line with the main idea and contributions. Can be improved.

The table-1 need to be corrected, because the challenges and its content is not referred with any reference research paper further its relevancy with the edge paradigm is not established.

--   Authors did not highlight the objectives of their research.

-  Authors did not highlight their contributions.

-  Gartner states that 30% of industrial control systems are expected to have analytics and AI edge reasoning capabilities by 2025. Considering the importance of this rise the authors mentioned AI + IoT Edge Computing as an important component. Similarly, a recent study (a) refers to the using artificial intelligence in general and machine learning either deep models or lightweight models for offloading decisions optimizations. Authors must take advantage of this study and include the relevant stuff.

- Moreover, in section 5.2 mobility is not discussed in detail that’s the need of today’s Vehicular Edge Networks. The study (b) can help in this. The authors should include relevant information from it.

- In addition, the user speed and direction has also a large impact on system latency in edge computing paradigm,; the study (b) show  quite interesting findings over it that should be included in section 5.2. The authors should add these studies in their paper and describe the impact of AI on offloading and VM migration scenarios specifically to optimize the offloading decisions.

(a). Mobility-Aware Computational Offloading in Mobile Edge Networks: A Survey

(b). COME-UP: Computation Offloading in Mobile Edge Computing with LSTM Based User Direction Prediction

-   The sub-section 2.2 needs to be explained with the proper algorithm-based justification for the decision taken at end edge, Net edge, and Cloud edge.

-     The entire paper's English should be double-checked and improved.

The content caching in IoT based EDGE Computing architecture is very critical challenge for modern applications. The authors should add a section in it and include some recent ideas on it such as,

CoPUP: content popularity and user preferences aware content caching framework in mobile edge computing

-   The authors should include a section at the end summarizing the future research directions and some open challenges in IoT-Edge Computing.

-   Energy consumption is also very critical parameters in edge computing..  The authors can add some details or add a subsection on it describing the impact of energy on IoT, Smart Grid, Smart City, Telematics, Smart Home`s echo system.
